



# Uncertainty in fault seal parameters: implications for CO₂ column height retention and storage capacity in geological CO₂ storage projects

Johannes M. Miocic[1], Gareth Johnson[2], Clare E. Bond[3]

[1]Institute of Earth and Environmental Sciences, University of Freiburg, Albertstr. 23b, 792104 Freiburg, Germany

[2] Department of Civil and Environmental Engineering, University of Strathclyde, James Weir Building, Glasgow, G1 1XJ, UK

[3] School of Geosciences, Department of Geology and Petroleum Geology, Meston Building, Aberdeen University, Aberdeen AB24 3UE, UK

*Correspondence to*: Johannes M. Miocic (johannes.miocic@geologie.uni-freiburg.de)

**Abstract.** Faults can act as barriers to fluid flow in sedimentary basins, hindering the migration of buoyant fluids in the subsurface, trapping them in reservoirs and facilitating the build-up of vertical fluid columns. The maximum height of these columns is reliant on the retention potential of the sealing fault with regards to the trapped fluid. Several different approaches for the calculation of maximum supported column height exist for hydrocarbon systems. Here, we translate these approaches to the trapping of carbon dioxide by faults and asses the impact of uncertainties in i) the wettability properties of the fault rock, ii) fault rock composition, and iii) reservoir depth, on retention potential. In similarity to hydrocarbon systems, uncertainties associated with the wettability of a $CO_2$-brine-fault rock system for a given reservoir have less of an impact on column heights than uncertainties of fault rock composition. However, the wettability of the carbon dioxide system is highly sensitive to depth, with a large variation in possible column height predicted at 1000m and 2000m depth, the likely depth range for carbon storage sites. In contrast to hydrocarbon systems higher phyllosilicate entrainment into the fault rock may reduce the amount of carbon dioxide that can be securely retained. Our results show that if approaches developed for fault seal in hydrocarbon systems are translated, without modification, to carbon dioxide systems the capacity of carbon storage sites will be inaccurate, and the predicted security of storage sites erroneous.

## 1 Introduction

Carbon capture and storage, is one of the key technologies to mitigate the emission of anthropogenic carbon dioxide ($CO_2$) to the atmosphere (IPCC, 2005; Benson and Cole, 2008; Haszeldine, 2009; Aminu et al., 2017). Fault seal behaviour will impact geological $CO_2$ storage security as well as storage capacity calculations. For the successful widespread implementation of CCS, the long-term security of storage sites is vital and the fate of injected $CO_2$ needs to be understood. Faults are of major importance as potential fluid pathways for both vertical and lateral migration of fluids in the subsurface (Bjørlykke, 1993;



Sibson, 1994; Bense et al., 2013). Assessing whether a fault forms a lateral flow barrier or baffle for $CO_2$ is crucial to assessing the efficiency and safety of sub-surface carbon storage, as faults are ubiquitous in sedimentary basins, the most likely $CO_2$ storage reservoirs, and will naturally occur close to or within storage complexes. Indeed, faults occur at many of the first industrial and pilot scale $CO_2$ storage sites located in sedimentary basins (e.g. In Salah, Algeria (Mathieson et al., 2010);

Snøhvit, Norway (Chiaramonte et al., 2011); Ketzin, Germany (Martens et al., 2012); Otway, Australia (Hortle et al., 2013)).

Faults influence the flow and migration of fluids in three basic ways: (i) They can modify flow paths by juxtaposing stratigraphically distinct permeable and impermeable units against each other (Fig. 1a, Allan, 1989). (ii) The petrophysical properties of fault rocks can impede cross-fault flow between permeable units (Fig. 1b, Yielding et al., 1997; Aydin and Eyal,

2002; van der Zee and Urai, 2005) and (iii) faults can provide fault parallel flow between separate permeable units (Fig. 1c, (Eichhubl et al., 2009; Dockrill and Shipton, 2010). i) assumes no (or minimal) permeability change in the fault zone, whereas ii) and iii) require permeability reduction and increase respectively. For $CO_2$ storage sites the latter two mechanisms are of particular interest and are considered here. It is worth noting that these permeability changes are temporal and dynamic and fault reactivation (Barton et al., 1995; Wiprut and Zoback, 2000) should be an important consideration in $CO_2$ storage projects.

Whether a fault is sealing or non-sealing is dependent on the structure and composition of the rock volume affected by, and the mechanics of, faulting (Caine et al., 1996; Aydin, 2000; Annunziatellis et al., 2008; Faulkner et al., 2010). Caine et al. (1996) describe fault zones in siliciclastic rocks defined by a fault slip surface and core and an associated damage zone, and considered the changes in permeability of a fault in this context. Fault damage zones and the fault cores are interpreted as having contrasting mechanical and hydraulic properties with the fault core being often rich in phyllosilicates which typically

have low permeability while open fractures in the damage zone can have a substantially higher permeability than the host rock (Caine et al., 1996; Faulkner and Rutter, 2001; Guglielmi et al., 2008; Cappa, 2009). Models for fault zone characterisation have evolved and describe fault zones with single high-strain cores (Chester and Logan, 1986) and containing several cores (Faulkner et al., 2003), with cores and slip surfaces at the edge of the fault zone and in the middle. Perhaps to think of it simply; one models does not fit all and the heterogeneities in natural fault systems and rocks result in unique fault geometries and

evolutions, albeit with similarities and semi-predictable processes.
When a fluid lighter than the pore-filling brine, such as hydrocarbons or $CO_2$, is introduced into a reservoir, it will naturally migrate upwards due to the buoyancy effect until it encounters a flow barrier such as a caprock or potentially a fault. The fluid will accumulate underneath the flow barrier until breakthrough occurs due to the increase in pressure within the reservoir. The maximum vertical extent of the fluid underneath the seal before seal failure, often referred to as column height, is controlled

by the fluid flow properties of the seal with regards to the fluid (Wiprut and Zoback, 2002). In the hydrocarbon industry, column heights are routinely calculated as they estimate the maximum amount of oil or gas that could be accumulated within a prospect (Downey, 1984). As the fluid flow properties of the seal may vary spatially, some uncertainty is associated with column heights, in particular when faults with their associated heterogeneities form reservoir-bounding seals. In the context of





$CO_2$ storage, column heights represent the maximum amount of $CO_2$ that could be stored within a reservoir before migration out of the reservoir.

Evidence from outcrop studies indicate that faults play an important role for the migration of $CO_2$ in the subsurface. Both fault parallel migration of $CO_2$ in fault damage zones (Annunziatellis et al., 2008; Gilfillan et al., 2011; Kampman et al., 2012;
Burnside et al., 2013; Keating et al., 2013, 2014; Frery et al., 2015; Jung et al., 2015; Skurtveit et al., 2017; Bond et al., 2017; Miocic et al., 2019) and across-fault migration has been reported (Shipton et al., 2004; Dockrill and Shipton, 2010). Studies of natural analogues for $CO_2$ storage sites have shown that if naturally occurring $CO_2$ reservoirs fail to retain column heights of $CO_2$ in the subsurface this is almost exclusively due to fault leakage (Miocic et al., 2016; Roberts et al., 2017).

In this contribution we review the main methods used to predict hydrocarbon column heights for fault bound reservoirs as
applied to hydrocarbons. Placing these into a $CO_2$ context we consider the implications of the assumptions used and their applicability for $CO_2$ storage. Stochastic simulations are used to test the impact of $CO_2$ specific uncertainties on different fault seal algorithms, and how these affect the predicted $CO_2$ column height. The results highlight that fault seal parameters are poorly constrained for $CO_2$ and can significantly change the predicted $CO_2$ storage volume in fault-bounded reservoirs. Importantly, our results suggest that increasing amounts of phyllosilicates within the fault core, normally associated with
increasing fault impermeability, may not necessarily increase the $CO_2$ column height within a reservoir.

## 2 Predicting fault seals for hydrocarbons and implications for $CO_2$ storage

As they are lighter than the pore-filling brine, hydrocarbons (HCs) migrate to the top of a reservoir where they accumulate underneath a seal. The buoyancy of HCs creates a pressure difference of $\Delta P$ at the seal-reservoir interface that is proportional to the hydrocarbon plume/column height (h) and the difference in mass density between brine ($\rho_W$) and HC ($\rho_{hc}$):

$$\Delta P = (\rho_w - \rho_{HC})\, g\, h \tag{1}$$

where g = gravitational constant and the density of HCs is dependent on the phase (gas or oil) and the in-situ pressure and temperature conditions.

The transport of HCs within rocks is controlled by capillary forces: the interfacial tension (IFT) between HCs and the brine, the wettability of the rock/mineral surface (wetting or contact angle, $\theta$) with respect to HCs, and the structure (size) of the pore
system. Capillary pressure ($P_C$), the pressure difference that occurs at the interface of HCs and brine, is commonly expressed as:

$$P_c = P_{HC} - P_{brine} = \frac{2\,IFT \times \cos\theta}{r} \tag{2}$$

Where $P_{HC}$ is the pressure of the HC, $P_{brine}$ is the pressure of the brine and r is the pore throat radius. $P_c$ is inversely proportional to the pore throat radius and thus fine-grained rocks with small pores exhibit larger $P_c$ and act as flow barrier to migrating HCs
– leading to the accumulation of fluids underneath fine-grained rocks.



For HCs the wettability parameters IFT and θ are fairly constant at subsurface reservoir conditions, with IFT of oil decreasing from around 25 mN/cm at very shallow conditions to around 10mN/cm for conditions commonly found in reservoirs at 2.5 km depth (Schowalter, 1979; Watts, 1987). For methane IFT is around 25 mN/cm at subsurface conditions (Schowalter, 1979). The contact angle for HCs is commonly reported as 0°(Vavra et al., 1992), simplifying equation 2 as the cosine of 0° is 1.

However, for other fluids such as $CO_2$ the wettability parameters IFT and θ are strongly pressure and temperature dependent. Due to the heterogeneous nature of rocks the size of pores within the sealing rock (fault rock or cap rock) varies and thus two capillary pressures can be defined. Firstly, the capillary entry pressure ($P_e$), which controls the initial intrusion of the non-wetting fluid into the low permeability rock and is controlled by the radius of the largest pore throat that is in contact with the reservoir rock. Secondly, and of higher interest to column height calculations, the capillary threshold pressure ($P_{th}$) or

sometimes called capillary breakthrough pressure, at which the wetting phase in the low permeability rock is displaced to an extent that the percolation threshold is exceeded and a continuous flow path of the non-wetting phase forms across the pore-network. The capillary threshold pressure is controlled by the smallest pore throat along the flow path, and thus $P_e < P_{th}$ applies. Seal failure occurs when buoyancy pressure is larger than capillary breakthrough pressure and the maximum supported column height follows from equations 1 and 2:

$$h = \frac{2\,IFT \times \cos\theta}{r} \times \frac{1}{(\rho_w - \rho_{HC}) \times g}$$ (3)

The ability of fault bound reservoirs to retain significant column heights thus depends on the fault rock composition which controls the pore-throat size (r) as well as the wettability parameters (IFT, θ). The composition and type of fault rocks in siliciclastic rocks is mainly influenced by (i) the composition of the wall rocks that are slipping past each other at the fault and in particular their content of fine grained phyllosilicate clay minerals, (ii) the stress conditions at the time of faulting, and (iii)

the maximum temperature that occurred in the fault zone after faulting (Yielding et al., 2010).

In clay poor sequences (i.e. clean sandstones with less than 15% clay), the dominant fault rock types are disaggregation zones and catalases (Fisher and Knipe, 1998; Sperrevik et al., 2002). Disaggregation zones form during fault slip at low confining stress during early burial and constitute grain reorganization without grain fracturing. Thus they tend to have similar hydraulic properties as their host sandstones and do not form flow barriers (Fisher and Knipe, 2001). At deeper burial (> 1 km) and

higher confining stresses cataclastic processes are more significant and the resulting fractured grain fragments block the pore space resulting in higher $P_{th}$ and with permeabilities on average one to two magnitudes lower than the host rock (Fisher and Knipe, 2001). Additionally, quartz cementation can further lower permeabilities in both disaggregation zones and cataclasites if they are subjected to post-deformation temperatures of >90°C, which equates to >3 km burial depths at typical geothermal gradients (Fisher et al., 2000).

In sequences with intermediate clay content (15-40 % phyllosilicate), fault rocks are formed by a deformation-induced mixing of generally unfractured quartz grains and clay matrix. The resulting texture creates a fault rock with a texture termed clay-matrix gouge or phyllosilicate framework fault rock (Fisher and Knipe, 1998). Due to the clay content these fault rocks generally have high $P_{th}$ and low permeabilities (Gibson, 1998).



In sequences dominated by clay or shale beds (>40% phyllosilicate), clay and shale rich smears can be formed on the fault plane (Weber et al., 1978). Such smears occur during ductile deformation at depths where the beds are not strongly consolidated and are often wedge-shaped, with the thickest smear adjacent to the source bed (Aydin and Eyal, 2002; Vrolijk et al., 2016).

If faulting occurs at deeper burial depths where the beds are lithified, shale smears can be generated by abrasional rather than ductile processes. In such cases thin shale coatings of more or less constant thickness are formed along the fault plane (Lindsay et al., 1993). Gaps within the clay and shale smears can occur at any point (Childs et al., 2007), lowering the hydrocarbon sealing capacity of the fault rock significantly.

As direct information on fault rock composition is very rare for subsurface cases, several algorithms have been developed in the past decades to estimate the probable fault rock composition at each point of the fault surface (Weber et al., 1978; Fulljames et al., 1997; Lindsay et al., 1993). The widely used Shale Gouge Ratio (SGR) algorithm takes the average clay content of beds that slipped past any point (based on fault throw) (Yielding et al. (1997)):

$$SGR = \frac{\sum(\text{Clay content} \times \text{bed thickness})}{\text{throw}} \times 100\% \qquad (4)$$

SGR can be used as an estimate of fault rock composition, with high SGRs (>40-50%) the fault rock is assumed to be dominated by clay smears, while low SGRs (<15-20%) indicate that the fault rock is likely to be disaggregation zones or cataclasites (Yielding et al., 2010). The SGR algorithm, similar to other algorithms like the Shale Smear Factor (Lindsay et al., 1993), the Clay Smear Potential (Fulljames et al., 1997) or the Probabilistic Shale Smear Factor (Childs et al., 2007) which all use a combination of throw clay bed distribution or thickness to predict the effects of clay smears, do not consider the detailed fault

rock distribution and fault zone complexity observed on outcrops or at the centimetre, and sub-centimetre scale (Faulkner et al., 2010; Schmatz et al., 2010). It has however been successfully used during the last two decades to predict hydrocarbon fault seals in the subsurface (Manzocchi et al., 2010; Yielding, 2012).

Two different approaches to link SGR and fault rock composition estimation with fault seal prediction parameters such as

capillary threshold pressure have been developed over the years: (1) using known sealing faults to constrain relationships between SGR and HC column height and/or across fault pressure differences (Bretan et al., 2003; Yielding et al., 2010), and (2) measuring capillary threshold pressures and clay content of micro faults and correlating these to SGR, assuming that SGR is equivalent to the clay content of the fault rock (Sperrevik et al., 2002). The first approach has been fine-tuned with datasets from sedimentary basins around the world, while equations linking capillary pressure and clay content in the second approach

are derived from best-fit relationships of samples mainly from the North Sea:

$$P_{thB} = 10^{\left(\frac{SGR}{27} - C\right)} \; (Bretan\ et\ al., 2003) \qquad (5)$$

with C = 0.5 for burial depths of less than 3 km, C= 0.25 for burial depths of 3.0-3.5 km and C=0 for burial depth greater than 3.5 km.



$$P_{thY} = 0.3 \times SGR - 6 \quad (Yielding, 2012) \tag{6}$$

(for burial depth of less than 3 km) and

$$P_{thY} = 0.15 \times SGR + 1.9 \quad (Yielding, 2012) \tag{7}$$

for burial depths of more than 3.5km, and

$$P_{thS} = 31.838 \times k_f^{-0.3848} \quad (Sperrevik\ et\ al., 2002) \tag{8}$$

where $P_{thS}$ is the Hg-air fault rock threshold pressure and $k_f$ the fault rock permeability:

$$k_f = 80000 \exp\{-[19.4 \times SGR + 0.00403\ z_{max} + (0.0055\ z_f - 12.5)(1 - SGR)^7 \tag{9}$$

where $z_{max}$ is the maximum burial depth and $z_f$ is the depth at the time of faulting.

These three algorithms (Eq. 5-9) are widely applied to predict fault seals. In combination with equation 3 they can be used to calculate maximum fluid-column heights. While the Bretan et al. (2003) algorithm (Eq. 5) assumes an exponential correlation between the fault rock clay content (FRCC) and the capillary threshold pressure, Yielding's (2012) algorithm (Eq. 6 & 7) is based on the assumption of a linear correlation between these variables. The Sperrevik et al (2002) (Eq. 8 & 9) algorithm also assumes an exponential relationship, but tends to predict lower capillary threshold pressures than the Bretan et al (2003)

algorithm (Fig. 3). Note that reported capillary pressures are sometimes measured in Hg-air-rock systems, which are often used to experimentally derive capillary pressures. In order to convert them to fluid-brine-rock systems, the following equation is used:

$$P_{HC-brine} = P_{Hg-air} \times \frac{IFT_{HC-brine} \times \cos\theta_{HC-brine}}{IFT_{Hg-air} \times \cos\theta_{Hg-air}} \tag{10}$$

Where P is capillary pressure, IFT interfacial tension and $\theta$ contact angle, indices indicate the fluid system. This equation

highlights that uncertainties of the wettability parameters can strongly influence capillary breakthrough pressures derived from mercury injection experiments (Heath et al., 2012; Lahann et al., 2014; Busch and Amann-Hildenbrand, 2013). Thus, the results of the three algorithms are not necessarily directly comparable. Here we apply these equations (Eq. 5-10) to a $CO_2$ storage framework testing their veracity and analysing the revealed associated uncertainties.

**3 Fault seal algorithms for $CO_2$**

In contrast to the HC-brine-rock system, the wettability of the $CO_2$-brine-rock system is strongly controlled by temperature, pressure, and mineralogy (Iglauer et al., 2015b; Zhou et al., 2017). As a result, a fault seal that supports a certain hydrocarbon column height may not necessarily support a similar amount of $CO_2$ (Naylor et al., 2011). This highlights the need to have a good understanding of the $CO_2$ wettability in the subsurface in order to establish the security of carbon storage sites.

The IFT of the $CO_2$-brine system is temperature, pressure and salinity dependent. It decreases from ~72 to 25 mN/m as pressure increases from atmospheric to 6.4 MPa, and plateaus at around 25±5 mN/m for supercritical $CO_2$ conditions and deionized

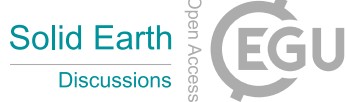

water (Kvamme et al., 2007; Chiquet et al., 2007; Espinoza and Santamarina, 2010). High salinity levels, as often found in the brines filling deep saline formations, can increase the interfacial tension by up to 10 mN/m (Espinoza and Santamarina, 2010; Saraji et al., 2014). Additionally, $CO_2$ dissolved in the brine may decrease IFT (Nomeli and Riaz, 2017), as may impurities such as $CH_4$ or $SO_2$ (Ren et al., 2000; Saraji et al., 2014). Thus for conditions most likely for storage reservoirs – supercritical

$CO_2$ at depths higher than 1200 m with saline brine (Miocic et al., 2016) - $CO_2$-brine IFT will be in the order of 35±5 mN/m (Fig. 4), similar to the range recently illustrated by Iglauer (2018).

The contact angle formed by the $CO_2$-brine interface on mineral surfaces varies strongly and is dependent on pressure and temperature conditions, mineral type, presence of organic matter, and the wetting phase (Sarmadivaleh et al., 2015; Espinoza and Santamarina, 2017). On water wet minerals, the contact angle ($\theta$) is about 40° on amorphous silica and calcite surfaces, $\theta$

~ 40° to 85° on mica, $\theta$ ~ 50° to 120° on coal, and $\theta$ ~ 8° to 30° on organic shale surfaces, while on oil wet amorphous silica $\theta$ ~ 85° to 95° (Chi et al., 1988; Chiquet et al., 2007; Chalbaud et al., 2009; Espinoza and Santamarina, 2010; Iglauer et al., 2015b; Arif et al., 2016; Espinoza and Santamarina, 2017; Guiltinan et al., 2017). With pressure rising from 10 to 15 MPa, $\theta$ increases up to 10° on quartz surfaces, while an increase of temperature from 50°C to 70°C at 10 MPa leads to an increase in $\theta$ of 15°(Sarmadivaleh et al., 2015). The $CO_2$ state also seems to influence the contact angle in oil-wet pores with $\theta_{gas} < \theta_{sc}$ (Li

and Fan, 2015). Additionally, the wettability of rocks may shift towards more hydrophobic the longer it is exposed to a $CO_2$-brine mixture (Wang and Tokunaga, 2015). From the experimental data available for conditions most likely for storage reservoirs, $\theta$ in water-wet conditions will range from ~40° for quartz dominated rocks to ~70° for an organic mica rich rock (Fig. 4), with higher values likely for deeper reservoirs (Iglauer, 2018).

A general issue with the wettability data available is that most experiments are done on single, very pure and cleaned mineral

surfaces and that data on the wettability of "real" subsurface rock-brine-$CO_2$ systems is very limited. Indeed, for potential caprock and reservoir rock lithologies such as dolomite, anhydrite, halite, mudrock, clays or fault rocks no data for subsurface conditions exists (Iglauer et al., 2015b). Recent developments for characterizing micro-scale variations of wettability in low permeability rocks may improve the knowledge in the future (Deglint et al., 2017). The wettability of fault rocks has to our knowledge not been studied experimentally yet but, as illustrated by the influence of mineralogy on contact angles, will depend

on fault rock composition.

As a wide range of IFT and CA values seem possible at the $CO_2$ – seal interface at the subsurface conditions likely for carbon storage sites, the sealing potential of faults for $CO_2$ and the conditions under which faults will form seals to $CO_2$ flow is unclear.

## 4 Markov Chain Monte Carlo modelling of fault seals for $CO_2$

In order to better understand the impact of the uncertainties of interfacial tension and contact angle (wettability) and fault rock

clay content (FRCC) described on commonly used fault seal algorithms when applied to $CO_2$, we run stochastic models where the input parameters follow probability distributions (i.e. have uncertainties associated). We use a Markov Chain Monte Carlo (MCMO) approach, which samples probability distributions of input parameters (Gilks et al., 1996), to statistically analyse the



effect of uncertainties in wettability and fault rock clay content (based on SGR) on the amount of $CO_2$ that can be securely stored in a fault bound reservoir. The input parameters are derived from the published data described: empirical values from Iglauer (2018) and experimental from (Botto et al., 2017; Iglauer et al., 2015b; Saraji et al., 2014). These parameters follow a normal distribution described by the mean and the standard deviation (σ) as seen in table 1 and are sampled randomly 20,000

times for each model run. Capillary threshold pressures for fault seals are calculated by using equations 5 to 9, these are then converted to the $CO_2$-brine system using equation 10, and subsequently column heights are calculated (eq. 3). The resulting column heights also follow a probability distribution (table 2).

Two theoretical cases are modelled: Reservoir A is located at 1000 m depth with a temperature of 45°C, a pressure of 10.2 MPa, and with a resultant $CO_2$ density of 515 Kg/m³. Reservoir B is located at a depth of 1800 m, has a temperature of 69°C,

a pressure of 18.36 MPa, and a resultant $CO_2$ density of 617 Kg/m³. Both reservoirs have a brine density of 1035 kg/m³, a maximum burial depth of 2000 m and a faulting depth of 1500 m. The normal distributions of the input parameters (FRCC (SGR) and wettability of the fault rock (CA, IFT)) for the MCMO modelling are listed in table 1. IFTs of 38 mN/cm and 34 mN/cm, and CAs of 50° and 70° are used as mean for the MCMO models of reservoir A and reservoir B, respectively, based on the IFT-depth and CA-depth relationships of Iglauer (2018). For each of the reservoirs 27 models were run with 20,000

iterations each, nine models for each of approaches that link SGR to fault rock threshold pressure (eq. 5 to 9). Of these nine models three simulate varying uncertainties in CA and IFT of the fault rock (models Wet1 to Wet3), three have varying uncertainties in FRCC (models FRC1 to FRC3), and three models calculate column heights based on uncertainties of FRCC as well as fault rock wettability (models Comb1 to Comb3).

Three additional models investigate the impact different fault rock compositions (and associated uncertainties) have on

supported column heights for reservoir A, using equation 3. Model No. 55 simulates a quartz rich fault rock (95% of IFT within 38±2 mN/cm, 95% of CA within 40±5°), model No. 56 a quartz-phyllosilicate mixture (95% of IFT within 38±2 mN/cm, 95% of CA within 60±5°), and model 57 a phyllosilicate rich fault rock (95% of IFT within 35±2 mN/cm, 95% of CA within 75±5°).

The results of the MCMO models highlight the differences between the three approaches that link FRCC to fault rock threshold

pressure with the approach of Sperrevik et al. (2002) generally resulting in lower column heights the approaches of Bretan et al. (2003) and Yielding (2012) for both Reservoir A and B (Tab. 2, Figs. 5 & 6). Uncertainties in the wettability of fault rocks (CA, IFT) have less of an impact on the supported column height distributions than uncertainties in FRCC.

For reservoir A, the models which are used to investigate the impact of uncertainties in wettability (Wet1-Wet3) have column heights ranging from 14.8±0.9 m to 14.6±3.6 m (after Sperrevik et al., 2002), and from 73±4 m to 72±18 m (after Bretan et

al., 2003), and from 111±6 m to 110±27 m (after Yielding, 2012). Models which simulate uncertainties in FRCC in the same reservoir have column heights ranging from 16±7 m, and from 74±14 m to 95±80 m, and from 111±14 m to 111±55 m, for the three different approaches respectively. Models which combine the uncertainties of fault rock wettability and FRCC (Comb1-Comb3) have an even wider spread in column height distributions (Fig. 5c, f, i). For reservoir B, all models show a similar pattern to those of reservoir A (Fig. 6), however the mean supported column heights are only about 60% of those for reservoir



A due to the differences in fault rock wettability parameters (Tabs. 1, 2). This illustrates that conditions in deeper reservoirs may lead to smaller column heights.

The results of models 55 to 57 highlight that the composition of fault rocks strongly influences the supported column height. For conditions similar to reservoir A a quartz rich fault rock (model 55) can support a column height of 107±5 m, while a mixture of quartz and phyllosilicates is likely to support 70±6 m, and a phyllosilicate rich fault rock can only support a column height of 36±6 m.

## 5 Discussion

The results of the stochastic modelling show that uncertainties in fault rock wettability and FRCC can strongly influence the amount of $CO_2$ predicted to be securely stored within a fault-bound siliciclastic reservoir. While the uncertainty associated with the $CO_2$ wettability of fault and top seal rocks has implications for the geological storage of $CO_2$, the uncertainty of the composition of the fault rocks has a greater impact.

Use of SGR as a proxy for fault rock clay content is widely accepted and commonly used for hydrocarbon reservoirs (Fristad et al., 1997; Lyon et al., 2005). The algorithm linking SGR to fault zone threshold pressure/column height is a critical step in fault seal studies and our results show that different algorithms (Eq. 5-9) predict different $CO_2$ column heights. This is in line with other works comparing the three algorithms (Bretan, 2016) and is due to the sensitivity of the Sperrevik algorithm to the geological history (faulting depth and maximum burial). The algorithm has been developed from samples of North Sea cores from depths ranging from 2000-4500 m. The approaches by Bretan et al. (2003) and Yielding (2012) are both used to calculate the maximum threshold pressure, the approach by Sperrevik et al. (2002) gives an average threshold pressure. Thus, when used for a carbon storage capacity assessment, the column heights calculated with the algorithms of Bretan et al. (2003) and Yielding (2012) would illustrate the maximum potential storage capacity while the column heights resulting from the Sperrevik et al. (2002) algorithm would likely represent average capacities.

The high impact of SGR on column heights is predictable as SGR is a proxy for the amount of phyllosilicates that are incorporated into the fault rock and our results are in line with other work which highlight that good prediction of fault rock composition is crucial for hydrocarbon column height prediction (Fisher and Knipe, 2001; Yielding et al., 2010). When SGR is used for predicting fault seals in a hydrocarbon context higher SGR values coincide with higher contained column heights as high SGR value fault rocks have a higher phyllosilicate content. Contrastingly, our results show that for a $CO_2$ fluid a higher phyllosilicate content of the fault rock may result in lower column heights (Fig. 7) as the wettability of the $CO_2$ –brine-rock system is very different at subsurface conditions from the wettability of the HC-brine-rock system. Phyllosilicate minerals have contact angles of up to 85° while quartz has a contact angle around 40°(Espinoza and Santamarina, 2017; Iglauer et al., 2015a). Increasing the content of phyllosilicates in the fault rock (increasing FRCC and SGR) effectively increases the contact angle which directly reduces the capillary threshold pressure as the cosine of the contact angles approaches zero (Eq. 2). This indicates that an increase in phyllosilicates in the fault rock may not increase the amount $CO_2$ that can be retained by the fault



to the same degree as for hydrocarbons. This calls into question whether algorithms such as SGR, which assume that higher phyllosilicate content in fault gouges equal higher sealing properties, can be used to effectively predict $CO_2$ fault seals.

The results of our stochastic models also illustrate the impact of depth on the wettability of the $CO_2$-brine-rock system, with the deeper faulted reservoir scenario (at a depth of 1800 m) holding significantly lower column heights than the shallower

reservoir (depth of 1000 m). This is in contrast to fault seals for hydrocarbons where faults can retain higher fluid columns for similar SGR values in deeper reservoirs (Yielding, 2012). The influence of pressure on the sealing capacity of fault rocks for $CO_2$ has direct implications for the selection of carbon storage sites, with shallow reservoirs being able to retain a higher column of $CO_2$ than deeper reservoirs (Fig. 8). Note that minimum $CO_2$ storage site depths are around 1000 m and are governed by the $CO_2$ state and density (Miocic et al., 2016).

Non-sealing faults are often undesired in a hydrocarbon exploration context, this is not necessarily true in the case of carbon storage sites. Here, sealing faults may actually reduce the amount of $CO_2$ that can be safely stored within a reservoir as the lateral migration of the $CO_2$ plume is hindered and pressure build-up may occur (Chiaramonte et al., 2015; Vilarrasa et al., 2017). If fault rocks that are sealing for hydrocarbons are not necessarily sealing for $CO_2$, as the results of our study suggest, faulted abandoned hydrocarbon reservoirs could form good carbon storage sites as long as no vertical migration of $CO_2$ along

the fault occurs.

## 6 Conclusions

Fault seal modelling is associated with significant uncertainties, arising from the limited subsurface data, resolution of seismic data, faulting mechanics and fault zone structure, spatial and temporal variations, and overall limitations of scalability of observations. Nonetheless several models to estimate the sealing properties of faults have been developed and successfully

used to predict hydrocarbon column heights. However, for fault seal modelling of $CO_2$ reservoirs the wettability of the $CO_2$–brine-rock system introduces additional uncertainties and reduces the amount of $CO_2$ that can be securely stored within a reservoir compared to hydrocarbons.

In this study uncertainties in fault rock composition, as well as uncertainties of how $CO_2$ fluid-rock wettability properties of the reservoir change with depth, have a stronger impact on $CO_2$ column heights than uncertainties in wettability. Importantly,

a higher phyllosilicate content within the fault rock, which is commonly assumed to increase the threshold pressure, may reduce the threshold pressure due to increased $CO_2$-wetting behaviour with such minerals. In particular deep reservoirs/high pressures seem to lead to lower column heights when compared to the equivalent predicted hydrocarbon column height.

To ensure $CO_2$ storage security an appropriate site characterisation for storage sites is critical. Faults of all scales must be identified and their seal potential modelled with a range of uncertainties, including the fault rock composition and wettability.

During storage operations fault seal potential predictions could be refined by high resolution monitoring and development of databases similar to those used (Bretan et al., 2003; Yielding et al., 2010) to predicted hydrocarbon column heights. While

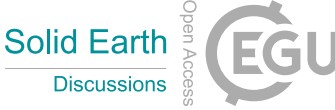



fault seals may impact the storage capacities it should be kept in mind that lateral migration through non-sealing faults can increase storage capacity.

**Code and data availability**

Model code is available from the corresponding author upon request.

**Author contributions**

JMM and GJ designed the project with input from CEB. JMM developed the model code and performed the MCMO simulations. The manuscript was written by JMM with contributions from both GJ and CEB.

**Acknowledgements**

This work was partly supported by the Panacea project (European Community's Seventh Framework Programme FP7/2007-2013, Grant No. 282900).

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





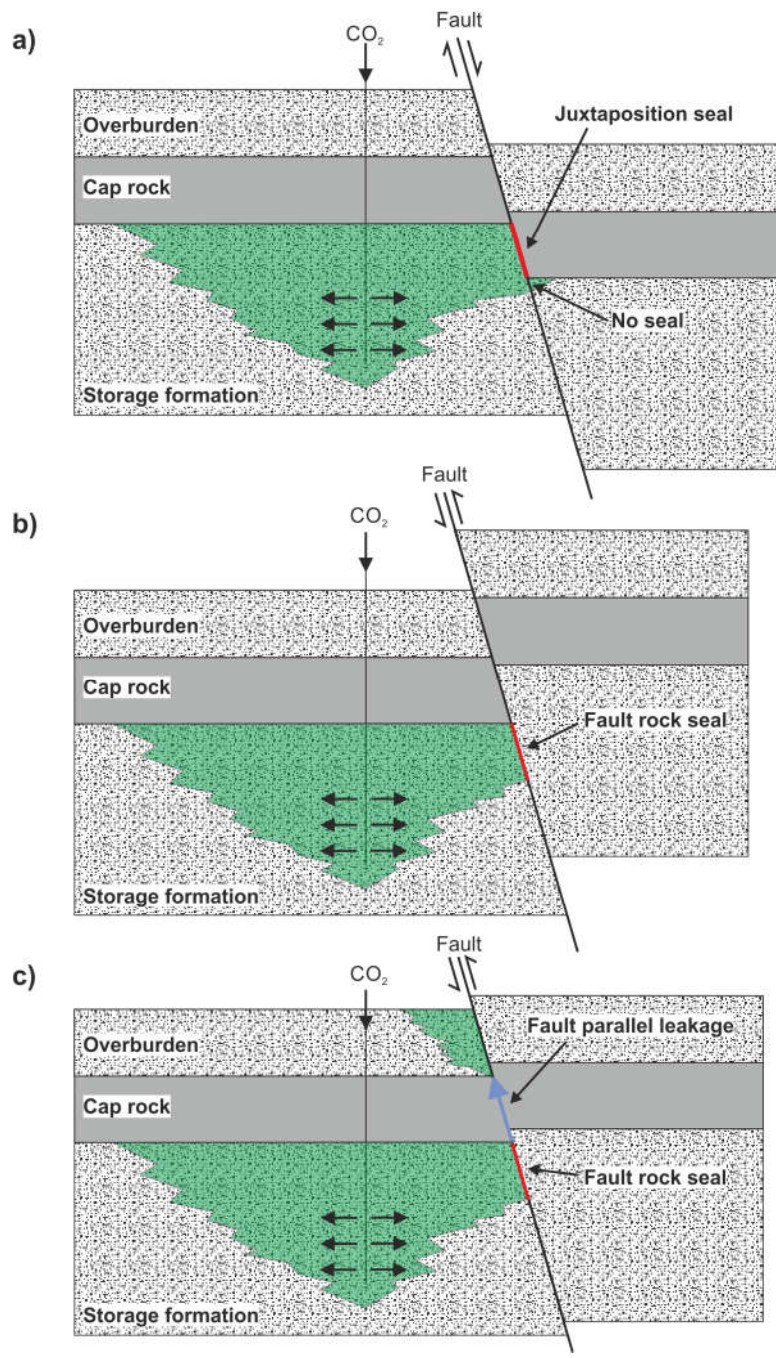

**Figure 1: Impact of faults on plume migration in a CO₂ storage site. a) Juxtaposition of the permeable storage formation and impermeable cap rocks, generating a juxtaposition seal. b) Impermeable fault rocks impede fluid flow within the storage formation (fault rock seal). c) Fault parallel, vertical migration bypasses the cap rock.**





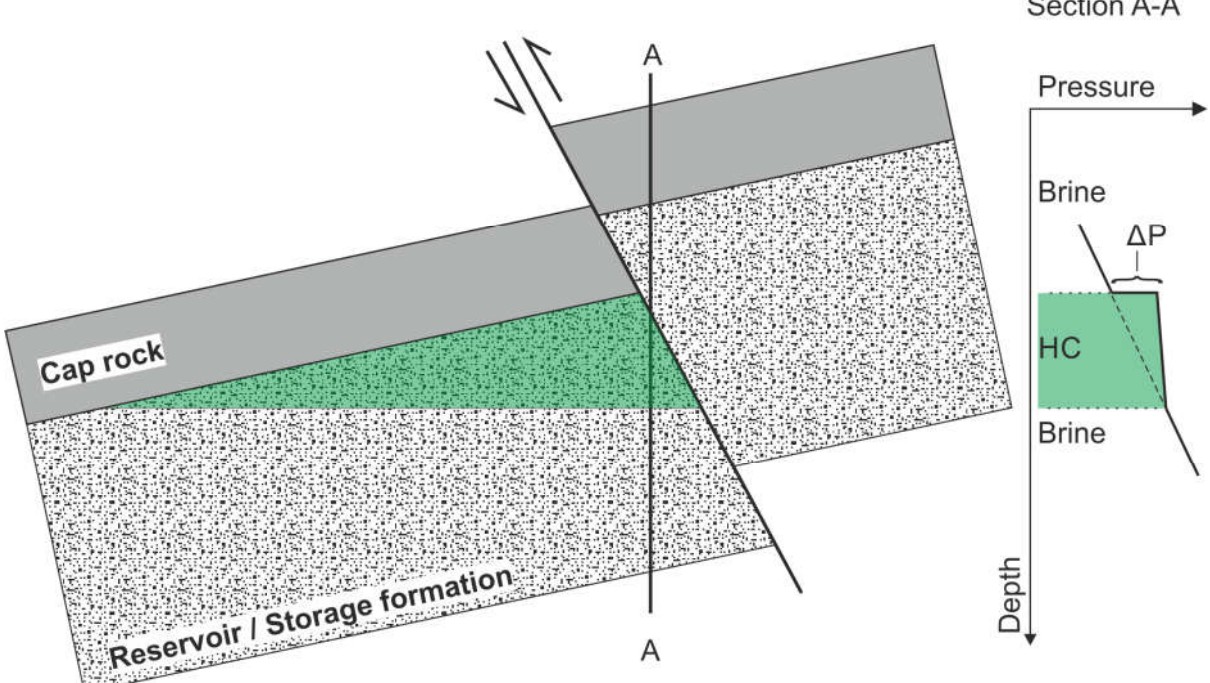

**Figure 2: Injection of CO2 into a faulted geological formation where the fault is sealing. The buoyancy of CO2 creates a pressure difference at the seal and fault displayed on a pressure/depth plot for the point of the diagram labelled A-A'.**





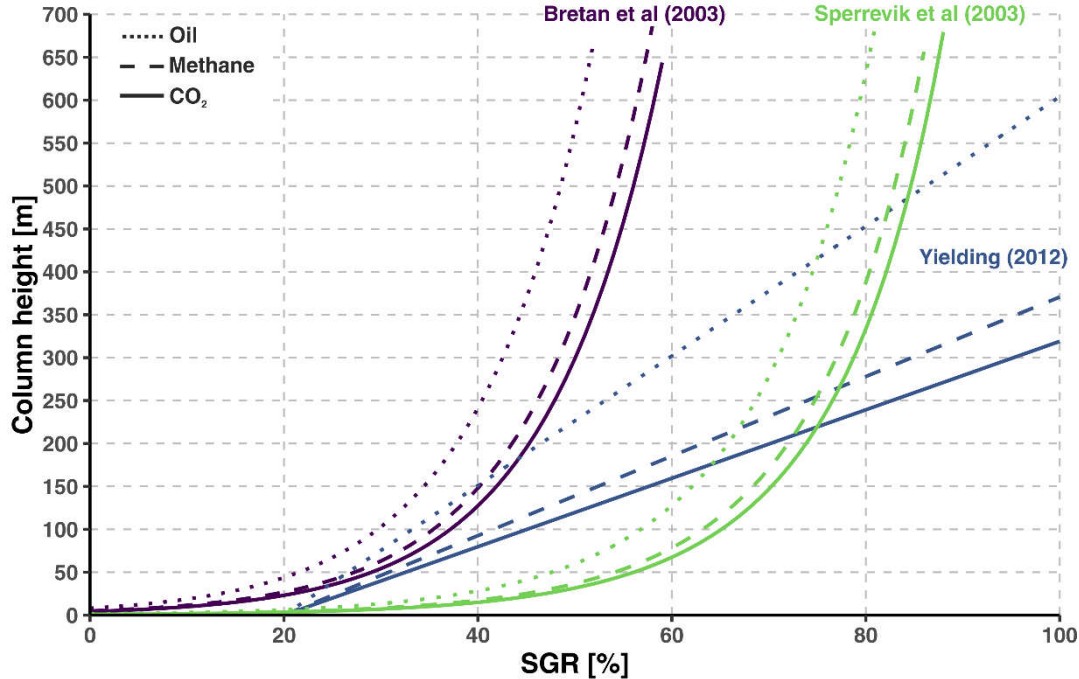

**Figure 3: Plot of SGR content of fault rocks and the resulting column heights for the algorithms of Bretan et al. (2003), Sperrevik et al. (2002) and Yielding (2012) for different fluid types for a reservoir at a depth of 1000 m. Assumes a contact angle of 50°and interfacial tension of 38 mN/m for the CO2-brine.rock system, a CO2 density of 515 kg/m³, a methane density of 75 kg/m³, an oil density of 700 kg/m³, and a brine density of 1,035 kg/m³.**



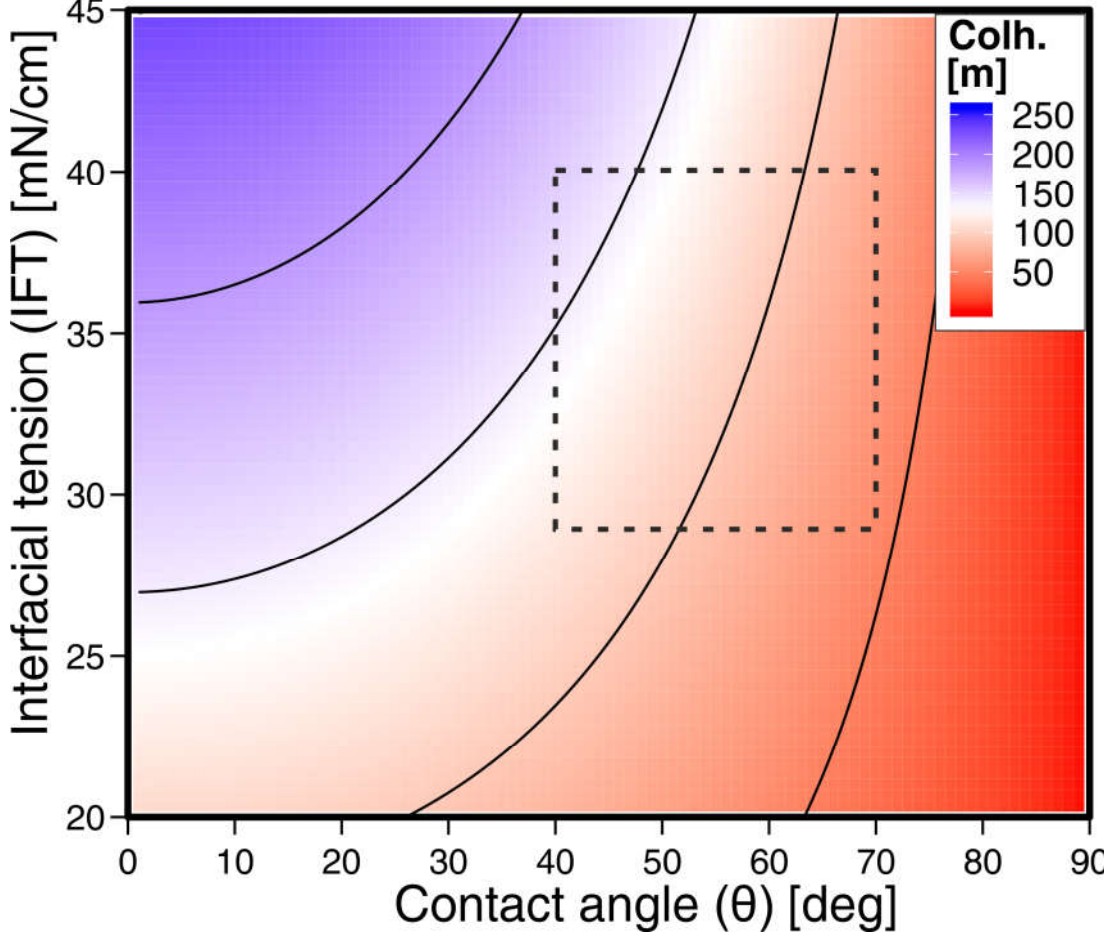

**Figure 4: Figure showing the influence of contact angle (θ) and interfacial tension (IFT) on supported CO₂ column height. Black lines are contours at 50 m intervals. The full range of IFT and θ shown here has been reported for CO₂-brine-rock systems, the dashed rectangle indicates conditions likely for geological storage. Column height calculated using equations 1 and 2 with a pore throat diameter of 100 nm, a typical value for organic-poor shales (Dong et al., 2017), and a CO₂ density of 630 kg/m³, correlating to a depth of about 1500 m.**







**Figure 5: Density distribution of column heights of models for reservoir A (models 1 to 27). The left column (A, D, G) illustrates the impact of uncertainties in fault rock wettability, the middle column (B, E, H) the impact of uncertainties in fault rock clay content (SGR), and the right column (C,F,I) the impact of combined uncertainties on column heights. Each row uses a different approach to link fault rock composition to threshold pressure. Uncertainty increases from dark to light coloured models (Tab.1). For all models N=20.000.**




**Figure 6: Density distribution of column heights of models for reservoir B (models 28 to 54). The left column (A, D, G) illustrates the impact of uncertainties in fault rock wettability, the middle column (B, E, H) the impact of uncertainties in fault rock clay content (SGR), and the right column (C,F,I) the impact of combined uncertainties on column heights. Each row uses a different approach to link fault rock composition to threshold pressure. Uncertainty increases from dark to light coloured models (Tab.1). For all models N=20.000.**





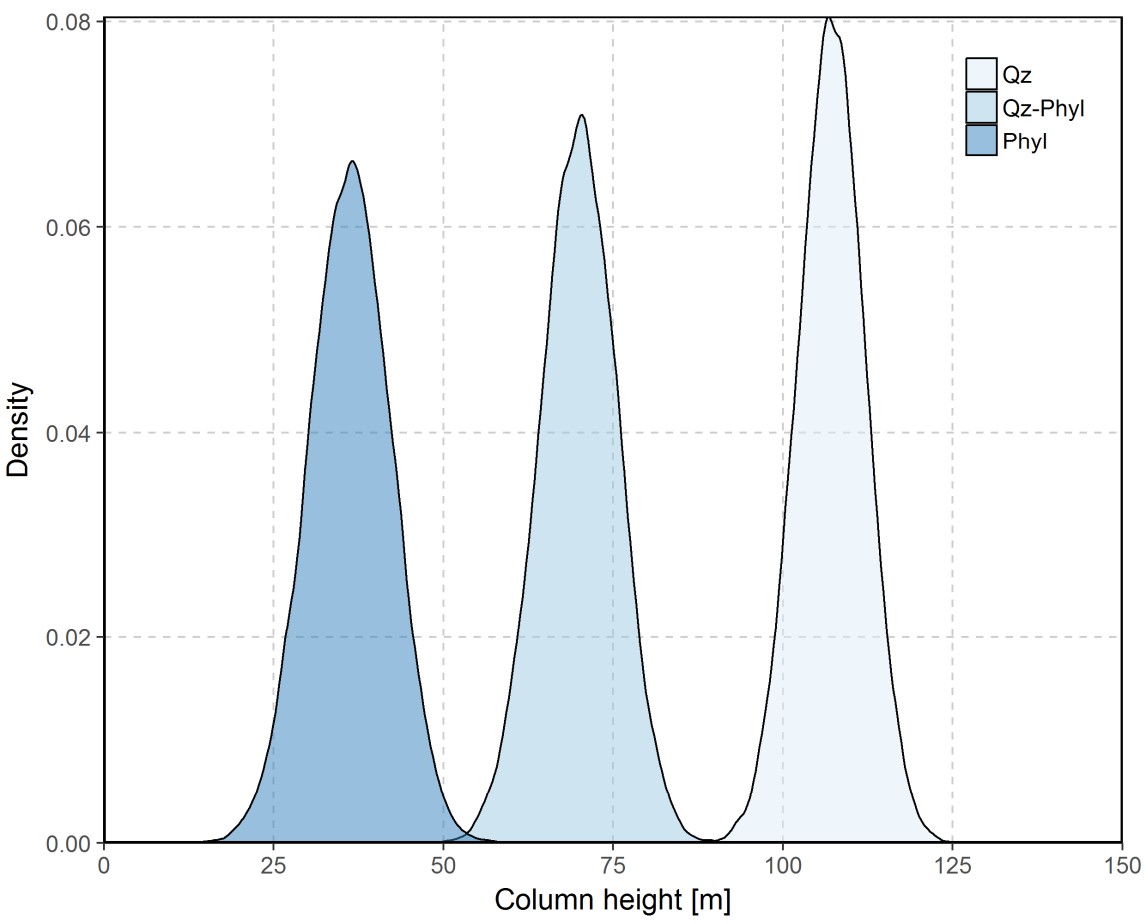

**Figure 7: Density distribution of column heights of models 55 to 57 illustrates the role of fault rock composition on supported column heights. Column height is calculated based on Eq.1 and Eq.2, for a pore throat size of 100 nm, and a $CO_2$ density of 515 kg/m³ (as Reservoir A). For all three models N=20.000.**



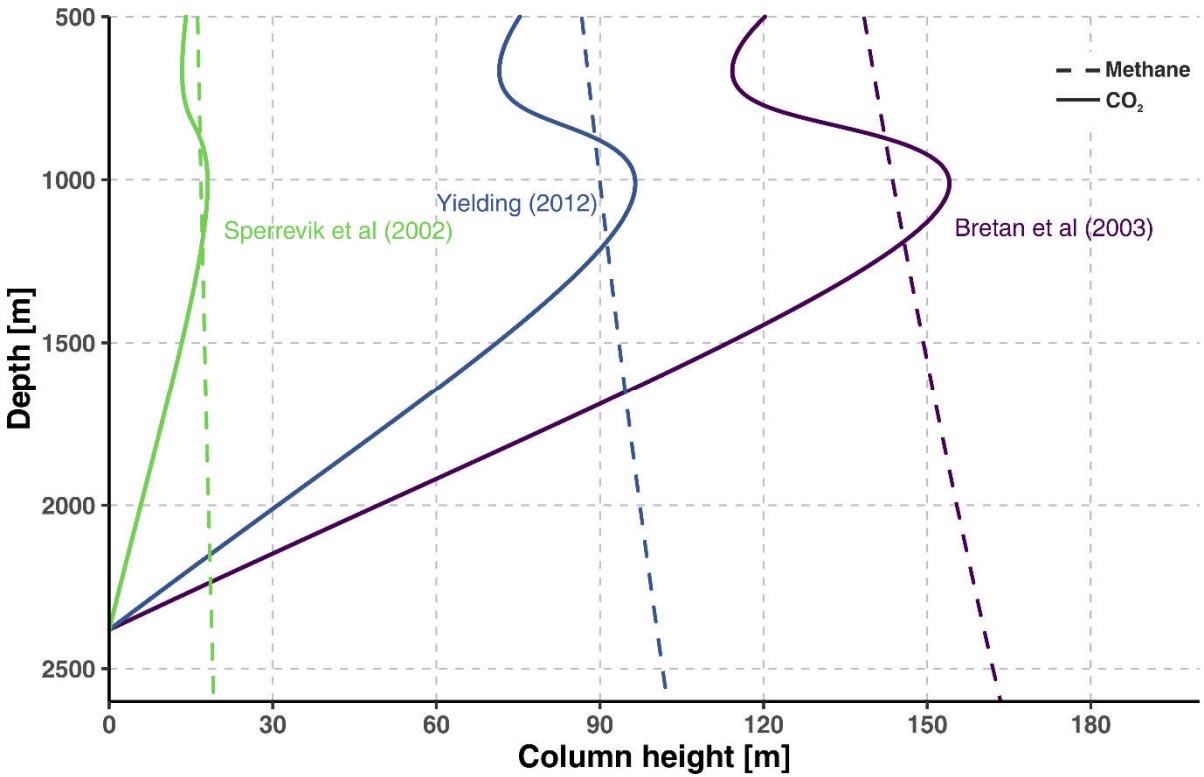

**Figure 8: Supported column heights of a fault with a phyllosilicate-rich fault rock (SGR=40) depending on the depth of the fault and the trapped fluid. For CO₂ the column height decreases with depth (after an optimum at ~1000 m depth) while methane column heights increase with depth. Based on depth-wettability relationships for CO₂ by Iglauer (2018).**

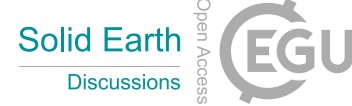



**Table 1: Table listing the input parameters for the MCMO modelling. Reservoir A and B refer to the two theoretical reservoirs described in the text, the approach refers to the algorithm used (see text), model indicates whether uncertainties in wettability parameters (Wet), fault rock composition (FRC) or combined uncertainties (Comb) are modelled. IFT is the interfacial tension, CA the contact angle and SGR the shale gouge ratio as parameter for fault rock composition. σ is the standard deviation and describes the shape of the input normal distribution.**

| Model No. | Reservoir | Approach | Model | IFT | σ | CA | σ | SGR | Σ |
|---|---|---|---|---|---|---|---|---|---|
| 1 | Reservoir A | Sperrevik et al. | Wet1 | 38 | 1 | 50 | 2.5 | 60 | |
| 2 | Reservoir A | Sperrevik et al. | Wet2 | 38 | 2.5 | 50 | 5 | 60 | |
| 3 | Reservoir A | Sperrevik et al. | Wet3 | 38 | 5 | 50 | 10 | 60 | |
| 4 | Reservoir A | Sperrevik et al. | FRC1 | 38 | | 50 | | 60 | 5 |
| 5 | Reservoir A | Sperrevik et al. | FRC2 | 38 | | 50 | | 60 | 10 |
| 6 | Reservoir A | Sperrevik et al. | FRC3 | 38 | | 50 | | 60 | 20 |
| 7 | Reservoir A | Sperrevik et al. | Comb1 | 38 | 1 | 50 | 2.5 | 60 | 5 |
| 8 | Reservoir A | Sperrevik et al. | Comb2 | 38 | 2.5 | 50 | 5 | 60 | 10 |
| 9 | Reservoir A | Sperrevik et al. | Comb3 | 38 | 5 | 50 | 10 | 60 | 20 |
| 10 | Reservoir A | Bretan et al. | Wet1 | 38 | 1 | 50 | 2.5 | 60 | |
| 11 | Reservoir A | Bretan et al. | Wet2 | 38 | 2.5 | 50 | 5 | 60 | |
| 12 | Reservoir A | Bretan et al. | Wet3 | 38 | 5 | 50 | 10 | 60 | |
| 13 | Reservoir A | Bretan et al. | FRC1 | 38 | | 50 | | 60 | 5 |
| 14 | Reservoir A | Bretan et al. | FRC2 | 38 | | 50 | | 60 | 10 |
| 15 | Reservoir A | Bretan et al. | FRC3 | 38 | | 50 | | 60 | 20 |
| 16 | Reservoir A | Bretan et al. | Comb1 | 38 | 1 | 50 | 2.5 | 60 | 5 |
| 17 | Reservoir A | Bretan et al. | Comb2 | 38 | 2.5 | 50 | 5 | 60 | 10 |
| 18 | Reservoir A | Bretan et al. | Comb3 | 38 | 5 | 50 | 10 | 60 | 20 |
| 19 | Reservoir A | Yielding | Wet1 | 38 | 1 | 50 | 2.5 | 60 | |
| 20 | Reservoir A | Yielding | Wet2 | 38 | 2.5 | 50 | 5 | 60 | |
| 21 | Reservoir A | Yielding | Wet3 | 38 | 5 | 50 | 10 | 60 | |
| 22 | Reservoir A | Yielding | FRC1 | 38 | | 50 | | 60 | 5 |
| 23 | Reservoir A | Yielding | FRC2 | 38 | | 50 | | 60 | 10 |
| 24 | Reservoir A | Yielding | FRC3 | 38 | | 50 | | 60 | 20 |
| 25 | Reservoir A | Yielding | Comb1 | 38 | 1 | 50 | 2.5 | 60 | 5 |
| 26 | Reservoir A | Yielding | Comb2 | 38 | 2.5 | 50 | 5 | 60 | 10 |
| 27 | Reservoir A | Yielding | Comb3 | 38 | 5 | 50 | 10 | 60 | 20 |
| 28 | Reservoir B | Sperrevik et al. | Wet1 | 34 | 1 | 70 | 2.5 | 60 | |
| 29 | Reservoir B | Sperrevik et al. | Wet2 | 34 | 2.5 | 70 | 5 | 60 | |
| 30 | Reservoir B | Sperrevik et al. | Wet3 | 34 | 5 | 70 | 10 | 60 | |
| 31 | Reservoir B | Sperrevik et al. | FRC1 | 34 | | 70 | | 60 | 5 |



| 32 | Reservoir B | Sperrevik et al. | FRC2 | 34 | | 70 | | 60 | 10 |
| 33 | Reservoir B | Sperrevik et al. | FRC3 | 34 | | 70 | | 60 | 20 |
| 34 | Reservoir B | Sperrevik et al. | Comb1 | 34 | 1 | 70 | 2.5 | 60 | 5 |
| 35 | Reservoir B | Sperrevik et al. | Comb2 | 34 | 2.5 | 70 | 5 | 60 | 10 |
| 36 | Reservoir B | Sperrevik et al. | Comb3 | 34 | 5 | 70 | 10 | 60 | 20 |
| 37 | Reservoir B | Bretan et al. | Wet1 | 34 | 1 | 70 | 2.5 | 60 | |
| 38 | Reservoir B | Bretan et al. | Wet2 | 34 | 2.5 | 70 | 5 | 60 | |
| 39 | Reservoir B | Bretan et al. | Wet3 | 34 | 5 | 70 | 10 | 60 | |
| 40 | Reservoir B | Bretan et al. | FRC1 | 34 | | 70 | | 60 | 5 |
| 41 | Reservoir B | Bretan et al. | FRC2 | 34 | | 70 | | 60 | 10 |
| 42 | Reservoir B | Bretan et al. | FRC3 | 34 | | 70 | | 60 | 20 |
| 43 | Reservoir B | Bretan et al. | Comb1 | 34 | 1 | 70 | 2.5 | 60 | 5 |
| 44 | Reservoir B | Bretan et al. | Comb2 | 34 | 2.5 | 70 | 5 | 60 | 10 |
| 45 | Reservoir B | Bretan et al. | Comb3 | 34 | 5 | 70 | 10 | 60 | 20 |
| 46 | Reservoir B | Yielding | Wet1 | 34 | 1 | 70 | 2.5 | 60 | |
| 47 | Reservoir B | Yielding | Wet2 | 34 | 2.5 | 70 | 5 | 60 | |
| 48 | Reservoir B | Yielding | Wet3 | 34 | 5 | 70 | 10 | 60 | |
| 49 | Reservoir B | Yielding | FRC1 | 34 | | 70 | | 60 | 5 |
| 50 | Reservoir B | Yielding | FRC2 | 34 | | 70 | | 60 | 10 |
| 51 | Reservoir B | Yielding | FRC3 | 34 | | 70 | | 60 | 20 |
| 52 | Reservoir B | Yielding | Comb1 | 34 | 1 | 70 | 2.5 | 60 | 5 |
| 53 | Reservoir B | Yielding | Comb2 | 34 | 2.5 | 70 | 5 | 60 | 10 |
| 54 | Reservoir B | Yielding | Comb3 | 34 | 5 | 70 | 10 | 60 | 20 |
| 55 | Reservoir A | | Qz | 38 | 1 | 40 | 2.5 | | |
| 56 | Reservoir A | | Qz-Phy | 38 | 1 | 60 | 2.5 | | |
| 57 | Reservoir A | | Phy | 38 | 1 | 75 | 2.5 | | |

**Table 2: Table showing the results of the MCMO models defined in table 1.**

| Model No. | Mean column height (m) | Standard deviation (m) | 2.5% percentile (m) | Median column height (m) | 97.5 % percentile (m) | N |
|---|---|---|---|---|---|---|
| 1 | 14.81 | 0.863 | 13.11 | 14.82 | 16.5 | 20000 |
| 2 | 14.78 | 1.821 | 11.21 | 14.78 | 18.37 | 20000 |
| 3 | 14.62 | 3.629 | 7.536 | 14.61 | 21.81 | 20000 |
| 4 | 16.15 | 6.946 | 6.886 | 14.82 | 33.29 | 20000 |
| 5 | 22.05 | 28.51 | 3.271 | 14.81 | 83.46 | 20000 |
| 6 | 1.23E+06 | 1.60E+08 | 0.7516 | 14.8 | 1154 | 20000 |





| | | | | | | |
|---|---|---|---|---|---|---|
| 7 | 16.1 | 7.071 | 6.755 | 14.68 | 33.45 | 20000 |
| 8 | 22.04 | 31.94 | 3.104 | 14.46 | 83.77 | 20000 |
| 9 | 3.38E+06 | 3.43E+08 | 0.6467 | 13.78 | 1087 | 20000 |
| 10 | 72.79 | 4.24 | 64.4 | 72.84 | 81.06 | 20000 |
| 11 | 72.61 | 8.945 | 55.08 | 72.61 | 90.24 | 20000 |
| 12 | 71.84 | 17.83 | 37.03 | 71.8 | 107.1 | 20000 |
| 13 | 73.98 | 13.81 | 50.6 | 72.81 | 104.3 | 20000 |
| 14 | 77.77 | 29.76 | 35.15 | 72.8 | 149.4 | 20000 |
| 15 | 95.2 | 80.8 | 16.97 | 72.77 | 306.6 | 20000 |
| 16 | 73.8 | 14.56 | 49.07 | 72.41 | 106.2 | 20000 |
| 17 | 77.3 | 31.59 | 32.76 | 71.5 | 154.6 | 20000 |
| 18 | 93.59 | 86.75 | 14.09 | 68.77 | 321.4 | 20000 |
| 19 | 111.5 | 6.494 | 98.62 | 111.5 | 124.1 | 20000 |
| 20 | 111.2 | 13.7 | 84.35 | 111.2 | 138.2 | 20000 |
| 21 | 110 | 27.31 | 56.7 | 110 | 164.1 | 20000 |
| 22 | 111.4 | 13.94 | 84.11 | 111.5 | 138.6 | 20000 |
| 23 | 111.3 | 27.88 | 56.7 | 111.5 | 165.6 | 20000 |
| 24 | 111.1 | 55.77 | 1.873 | 111.5 | 219.7 | 20000 |
| 25 | 111.2 | 15.5 | 81.36 | 110.8 | 142.6 | 20000 |
| 26 | 110.7 | 31.37 | 52.68 | 109.2 | 176 | 20000 |
| 27 | 109.1 | 63.19 | 1.101 | 103.2 | 247.4 | 20000 |
| 28 | 8.779 | 1.084 | 6.642 | 8.792 | 10.88 | 20000 |
| 29 | 8.761 | 2.2 | 4.468 | 8.769 | 13.11 | 20000 |
| 30 | 8.676 | 4.388 | 0.1707 | 8.671 | 17.44 | 20000 |
| 31 | 9.567 | 4.114 | 4.078 | 8.775 | 19.72 | 20000 |
| 32 | 13.06 | 16.88 | 1.938 | 8.772 | 49.43 | 20000 |
| 33 | 729600 | 9.45E+07 | 0.4452 | 8.765 | 683.7 | 20000 |
| 34 | 9.534 | 4.341 | 3.825 | 8.652 | 20.42 | 20000 |
| 35 | 13.05 | 19.5 | 1.624 | 8.37 | 51.46 | 20000 |
| 36 | 2.17E+06 | 2.27E+08 | 0.03721 | 7.316 | 641.1 | 20000 |
| 37 | 43.13 | 5.325 | 32.63 | 43.2 | 53.47 | 20000 |
| 38 | 43.05 | 10.81 | 21.95 | 43.09 | 64.41 | 20000 |
| 39 | 42.63 | 21.56 | 0.8387 | 42.6 | 85.69 | 20000 |
| 40 | 43.81 | 8.178 | 29.97 | 43.12 | 61.78 | 20000 |
| 41 | 46.06 | 17.63 | 20.82 | 43.12 | 88.49 | 20000 |
| 42 | 56.38 | 47.85 | 10.05 | 43.1 | 181.6 | 20000 |





| 43 | 43.7 | 9.91 | 27.24 | 42.77 | 65.86 | 20000 |
|----|------|------|-------|-------|-------|-------|
| 44 | 45.77 | 21.61 | 15.9 | 41.76 | 99.45 | 20000 |
| 45 | 55.43 | 60.6 | 0.5604 | 38.1 | 215.6 | 20000 |
| 46 | 66.05 | 8.155 | 49.98 | 66.16 | 81.88 | 20000 |
| 47 | 65.92 | 16.56 | 33.62 | 65.98 | 98.64 | 20000 |
| 48 | 65.28 | 33.02 | 1.284 | 65.24 | 131.2 | 20000 |
| 49 | 65.99 | 8.257 | 49.82 | 66.04 | 82.07 | 20000 |
| 50 | 65.92 | 16.51 | 33.58 | 66.04 | 98.1 | 20000 |
| 51 | 65.79 | 33.03 | 1.109 | 66.02 | 130.1 | 20000 |
| 52 | 65.84 | 11.71 | 44.45 | 65.38 | 90.08 | 20000 |
| 53 | 65.52 | 23.81 | 25.16 | 63.56 | 118 | 20000 |
| 54 | 64.57 | 49.28 | -6.554 | 56.9 | 180.7 | 20000 |
| 55 | 107.1 | 4.97 | 97.24 | 107.1 | 116.8 | 20000 |
| 56 | 69.93 | 5.649 | 58.82 | 69.98 | 80.98 | 20000 |
| 57 | 36.22 | 5.99 | 24.41 | 36.28 | 47.84 | 20000 |