# Peer review of "Uncertainty in fault seal parameters: implications for CO2 column height retention and storage capacity in geological CO2 storage projects"

_Solid Earth, 2019_

## Short Comment (SC1) · 10 Apr 2019

This paper makes a useful contribution to the discussion around the security of CO2 storage sites. Its findings that the fault seal calculations approaches conventionally applied to leakage of CO2 through faults that are based on those developed for hydrocarbon systems are actually not entirely suitable for CO2 should be modified. The author's findings that the influence of fault rock composition and depth (co2 phase) have a greater impact on fault seal calculations for CO2 leakage certainly should be accounted for. I have a few minor comments:

In your abstract your statement of your findings around line 15 is not as clearly stated

as it could be – especially with regards to your finding that fault composition is important and it may be worth moving your sentence beginning "In contrast to hydrocarbon systems higher phyllosilicate...". To before your sentence beginning "However, the wettability of the carbon dioxide system is highly sensitive..." only to make your point that increasing phyllosilicate may not result in the expected increase in fault sealing for a co2 system.

Page2 line 2 – while I agree that faults are ubiquitous in sedimentary basins, I would like to see a little more information provided on their extent, distribution, and scale, especially the impact of faults that are below seismic resolution. Page 2 line 10 – faults can provide fault parallel flow for impermeable units as well? fig 1c if you are suggesting flow through the fracture – if you are not suggesting fault flow then you may nbeed to reword this section and figure.

Page 2 line 28 – "will accumulate underneath the flow barrier until breakthrough occurs due to the increase in pressure within the reservoir" what do you mean by breakthrough? Capillary, spill point, induced fracturing? Can you be clearer with this statement? Page 3 line 30 – perhaps add the word seal before "rocks." Page 4 line 6 – you slightly contradict your statements at the end of page 3 where you infer fine grained rocks have small pore as page 4 states "Due to the heterogeneous nature of rocks the size of pores within the sealing rock (fault rock or cap rock) varies" Page 8 line 5 – I would remind the reader that equations 5 to 9 are the three different author approaches (Bretan etc) as that is how you refer to them in the subsequent modelling discussions. Discussion section – I would like to see a clearer statement of your findings and a more logical layout to your discussion as it is quite challenging to pick out the pertinent information- rather than saying it "strongly influences" – I would like to see clearer statement of what he influence such as increasing phyllosilicate fault rocks results in one third less co2 stored - to make your findings clear from the start of this section which can then lead onto the detailed discussion of the reasons why.

---

## Referee Comment (RC1) · Graham Yielding (Referee) · 27 Apr 2019

This manuscript raises important concerns regarding the applicability of conventional oil industry methods of fault seal analysis to the issue of CO2 trapping in underground storage facilities. The early sections provide a succinct review of the oil industry methods, highlighting the appropriate fluid properties which are relevant to the calculation of fault seal capacity (column height) for oil and gas and also CO2. The issues related to the wide variation in the CO2 experimental data are also well put. The Monte Carlo modelling technique used in this manuscript gives an interesting overview of the uncertainty in predicted column height as a result of uncertainty in the various input pa-

rameters. Emphasis is placed on how the different fluid properties of CO2 (as opposed to oil or methane) can lead to increased uncertainty.

There are some points which I believe need correction or clarification, in order to tighten the authors' arguments:

**In a number of places, starting at the top of p.4, the units of IFT are incorrectly stated as mN/cm instead of mN/m - the values quoted make it clear they should all be mN/m. This should be corrected throughout the ms.**

**Also at the top of page 4, I do not recognise the stated depth variability for hydrocarbon IFT. Firstly, IFT is not "fairly constant" but varies with depth. Oil IFT is stated to decrease with depth but in my experience it increases with depth - see Figure 9 of Yielding et al 2010, uploaded - from 25 mN/m at the surface to about 40 mN/m at around 3km. For methane, IFT decreases from around 75 mN/m at the surface, also to around 40 mN/m at 3km (see same figure).**

**On Figure 3, it is not clear to me how the CO2 lines have been derived for the two 'empirical' equations (Bretan & Yielding). I can see that the oil and methane lines differ appropriately because of their different density contrast with water. On density difference alone however, the CO2 line should be between oil and gas, so I assume the CO2 line is reduced because of either IFT or CA input or both (in a manner similar to eqn 10). However, both the Bretan & Yielding equations were based on agglomerated oil AND gas data (i.e. the data pool is blind to fluid type), so it is not clear to me what generic 'hydrocarbon' values for IFT & CA have been assumed, in order to compute the CO2 values via eqn 10. I request that these hydrocarbon IFT and CA values be added to the caption (already contains the CO2 values).**

**Related to the above point, the model results in D & G of Figs 5 & 6 report the effect of different ranges of wettability when using the Bretan & Yielding eqns. However, as you mention in the Discussion, these equations were derived as the maximum of observed data in oil & gas systems - so arguably they correspond to one extreme of**

the wettability range for hydrocarbons (least wettable). Depending on how that 'least wettable' state relates to your average CO2 wettability, I'm not sure if you are right to look at an uncertainty range which is symmetric about your chosen mean for CO2. Maybe I'm over-thinking this, perhaps you can offer a simple clarification.

**In the MC modelling, do you assume that all input parameters are independent? Is this valid? (E.g. do IFT and CA co-vary for CO2?).**

**In the MC modelling, I don't think it is mentioned in the text (or tables) what value of pore-throat size is used. A value of 100 nm is mentioned in the caption of Figure 7 but it isn't clear if this is used for all other models. This is an important point, because in general in oil industry fault seal analysis it is believed that the increasing seal capacity with increasing phyllosilicate content is overwhelmingly due to a corresponding decrease in pore-throat size. Pore-throat size is observed directly in microscope analysis and also back-calculated from Hg-air injection tests. Thus your results in Fig.7, showing smaller column height for phyllosilicate-rich fault rock than for qtz-rich fault rock because of changing CA, might be countered by the qtz-rich fault rock having larger pore throats. Maybe include this pore-throat variation in the input parameters to the Fig.7 models?**

The following are minor typos and points of expression:

**p1, line 15, should be "assess"**

**p1, line 16, "As with" might be better than "In similarity to"**

**p1, line 25, omit "," after storage**

**p2, line 23, replace ";" by ":"**

**p2, line 24, model not models**

**p2, line 27, omit "potentially"**

**p3, line 17, replace "lighter" by "less dense"**

**p3, line 23, replace "transport" by "trapping" (transport is controlled by permeability)**

**p4, line 9, replace "higher" by "greater"**

**p4, line 22, replace "catalases" by "cataclasites"**

**p4, line 24, insert "typically" before "> 1km"**

**p4, line 26, replace "with" by "in"**

**p5, line 24, insert "and" before "clay bed"**

**p5, line 24, replace "do" by "does"**

**p6, line 15, replace "sometimes" by "typically"**

**p7, line 30 onwards... I was confused by FRC (fault rock composition) and FRCC (fault rock clay content). I think you only need one of these, make it consistent in text and tables.**

**p9, line 3, insert "(see Figure 7)" after "57"**

**p9, line 26, insert "(and hence smaller pore-throat radius)" after "content" (see earlier comment)**

**Fig.4 - label the contours explicitly, to save the reader having to work out their values.**

**Table 1, last column. I assume the heading should be a lower case sigma, for standard deviation? The upper case symbol confused me for a while.**
* * *
[Figure]

**Fig. 9.** Some published estimates of the variation of
hydrocarbon–water interfacial tension with respect to
depth (pressure & temperature conditions). The methane
& decane curves indicate experimentally-measured
trends from Firoozabadi & Ramey (1988). The 'oil'
values are from Nordgård Bolås *et al.* (2005),
constructed from empirical equations of Firoozabadi &
Ramey (1988). Arrows show typical industry default
values for oil–water (green) and gas–water (red)
(d'Onfro, pers. comm., 2007).

**Fig. 1.**

---

## Author Comment (AC1) · 28 May 2019

Dear Graham Yielding,

We are grateful for your detailed review of our manuscript. The constructive suggestions and industry insights have been very helpful and have improved and strengthened the revised manuscript. In response to the reviews received we have made a number of changes which are outlined in the "response to reviewers" which has been uploaded as supplementary information along with the revised manuscript. We hope that the revisions sufficiently address the points raised.

[Figure]

Kind regards,

Johannes Miocic

---

## Author Comment (AC2) · 28 May 2019

Dear Katriona Edlmann,

We are grateful for your review of our manuscript. The constructive suggestions have been very helpful and have improved and strengthened the revised manuscript. In response to the reviews received we have made a number of changes which are outlined in the "response to reviewers" which has been uploaded as supplementary information along with the revised manuscript. We hope that the revisions sufficiently address the points raised.

[Figure]

Kind regards,

Johannes Miocic

---

## Author Comment (AC3) · 28 May 2019

**Title: "Uncertainty in fault seal parameters: implications for $CO_2$ column height retention and storage capacity in geological $CO_2$ storage projects"**

Our manuscript has benefited from two reviewers, Graham Yielding (R1) and Katriona Edlmann (R2). In the following we refer to the manuscript that was submitted for review as the "original manuscript", while the edited version which incorporates the reviewers comments is referred to as the "revised manuscript".

In the following the reviewer's comments are in *italic Arial* while our responses are in Times New Roman.

Author Comments to Reviewers for the open discussion.

To R1:

Dear Graham Yielding,

We are grateful for your detailed review of our manuscript. The constructive suggestions and industry insights have been very helpful and have improved and strengthened the revised manuscript. In response to the reviews received we have made a number of changes which are outlined in the "response to reviewers" which has been uploaded as supplementary information along with the revised manuscript. We hope that the revisions sufficiently address the points raised.

Kind regards,

Johannes Miocic

To R2:

Dear Katriona Edlmann,

We are grateful for your review of our manuscript. The constructive suggestions have been very helpful and have improved and strengthened the revised manuscript. In response to the reviews received we have made a number of changes which are outlined in the "response to reviewers" which has been uploaded as supplementary information along with the revised manuscript. We hope that the revisions sufficiently address the points raised.

Kind regards,

Johannes Miocic

**R1**

*In a number of places, starting at the top of p.4, the units of IFT are incorrectly stated as mN/cm instead of mN/m - the values quoted make it clear they should all be mN/m. This should be corrected throughout the ms.*

This has been adjusted in the revised manuscript.

*Also at the top of page 4, I do not recognise the stated depth variability for hydrocarbon IFT. Firstly, IFT is not "fairly constant" but varies with depth. Oil IFT is stated to decrease with depth but in my experience*

*it increases with depth - see Figure 9 of Yielding et al 2010, uploaded - from 25 mN/m at the surface to about 40 mN/m at around 3km. For methane, IFT decreases from around 75 mN/m at the surface, also to around 40 mN/m at 3km (see same figure).*

We thank the reviewer for their comment. In the original submission the depth variability of hydrocarbon

IFT was based on data from Schowalter (1979) and Watts (1987) and, as in the literature recommended by the reviewer, the IFT data in these publications does change with depth. The "fairly constant" statement of the original publication was supposed to illustrate that the IFT values for hydrocarbons are fairly constant at reservoir depths (1000-2000 m depth) when compared to $CO_2$ at these depths. However, we realise that this statement is misleading and have changed the section in the revised manuscript as follows:

*"For HCs the wettability parameters IFT and θ vary with depth, particularly large changes occur between surface and conditions found at depths of 1000 m.  IFT of oil increases from around 25 mN/m at very shallow conditions to around 40mN/m for conditions commonly found in reservoirs at 2.5 km depth (Yielding et al., 2010). For methane IFT is around 70 mN/m at surface conditions and decreases to 40*

*mN/m at subsurface conditions (Firoozabadi and Ramey, 1988; Watts, 1987). The contact angle for HCs is commonly reported as 0°(Vavra et al., 1992), simplifying equation 2 as the cosine of 0° is 1. However, for other fluids such as $CO_2$ the wettability parameters IFT and θ are even more pressure and temperature dependent."*

*On Figure 3, it is not clear to me how the CO2 lines have been derived for the two 'empirical' equations (Bretan & Yielding). I can see that the oil and methane lines differ appropriately because of their different density contrast with water. On density difference alone however, the CO2 line should be between oil and*

*gas, so I assume the CO2 line is reduced because of either IFT or CA input or both (in a manner similar to eqn 10). However, both the Bretan & Yielding equations were based on agglomerated oil AND gas*

*data (i.e. the data pool is blind to fluid type), so it is not clear to me what generic 'hydrocarbon' values for IFT & CA have been assumed, in order to compute the CO2 values via eqn 10. I request that these hydrocarbon IFT and CA values be added to the caption (already contains the CO2 values).*

The reviewer is correct in pointing this out. We have added the IFT and CA values to the figure caption.

Additionally, we have changed the assumed IFT values for both oil and methane in the calculation of the column heights to be in line with the values suggested by the reviewer in the previous comment (IFT for methane of 60 mN/m and 30 mN/m for oil). Note that for the resulting column heights the difference between methane and $CO_2$ columns is even larger.

*Related to the above point, the model results in D & G of Figs 5 & 6 report the effect of different ranges of wettability when using the Bretan & Yielding eqns. However, as you mention in the Discussion, these equations were derived as the maximum of observed data in oil & gas systems - so arguably they correspond to one extreme of the wettability range for hydrocarbons (least wettable). Depending on how that 'least wettable' state relates to your average CO2 wettability, I'm not sure if you are right to look at*

*an uncertainty range which is symmetric about your chosen mean for CO2. Maybe I'm over-thinking this, perhaps you can offer a simple clarification.*

We thank the reviewer for their comment. Indeed, when using the Bretan and Yielding equations the resulting column heights represent the maximum column heights instead of the average column heights resulting from the Sperrevik equations. Thus, where the Bretan and Yielding equations are used (Fig. 5&6 D&G) to analyse the influence of wettability uncertainty on column heights, we model around the least wettable state. Due to a lack of wettability data we use a normal distribution around this "least wettable" state. A more skewed distribution may better reflect the range in potential wettabilites, but this is not what we have considered here. We have clarified this in the revised manuscript by adding the following:
*"Note that equations 5 to 7 result in maximum column heights (or minimal wettability) while equations 8-9 give an average column height."*
And

*"IFTs of 38 mN/m and 34 mN/m, and CAs of 50° and 70° are used as mean wettability for the MCMO*
*models of reservoir A and reservoir B, respectively, based on the IFT-depth and CA-depth relationships of*
*Iglauer (2018). For models where the approaches by Bretan et al (2003) and Yielding (2010) are used,*
*these correspond to the mean least wettability."*

The comments by the reviewer offer some intriguing ideas about how future models could be developed
by using the whole input data of individual sealing faults from the Yielding and Bretan algorithms. These
could be used to calculate the distribution of buoyancy pressures for a given SGR and then translating the
buoyancy pressures to wettability (or pore throat size) distributions. If this could be combined with other
geological data, possibly new fault seal algorithms could be developed.

*In the MC modelling, do you assume that all input parameters are independent? Is this valid? (E.g. do*
*IFT and CA co-vary for CO2?).*

The reviewer points out an important point: theoretically IFT and CA should co-vary, with high IFT values
corresponding with low CAs and the other way around. However, for the $CO_2$-brine-rock system the
existing data in our opinion is too scattered to allow for a model in which they are dependent. We have
added text to the revised manuscript to highlight this point as follows:
*"The input parameters, which are all treated as independent, are derived from the published data*
*described:…"*

*In the MC modelling, I don't think it is mentioned in the text (or tables) what value of pore-throat size is*
*used. A value of 100 nm is mentioned in the caption of Figure 7 but it isn't clear if this is used for all other*
*models. This is an important point, because in general in oil industry fault seal analysis it is believed that*
*the increasing seal capacity with increasing phyllosilicate content is overwhelmingly due to a*
*corresponding decrease in pore-throat size. Pore-throat size is observed directly in microscope analysis*
*and also back-calculated from Hg-air injection tests. Thus your results in Fig.7, showing smaller column*
*height for phyllosilicate-rich fault rock than for qtz-rich fault rock because of changing CA, might be*
*countered by the qtz-rich fault rock having larger pore throats. Maybe include this pore-throat variation in*
*the input parameters to the Fig.7 models?*

We thank the reviewer for this comment, we have added the pore-throat size to the MC-modelling description in the revised manuscript. As suggested we have also added pore-size variations to the models 55-59 (two additional models) and these are shown in Figure 7. The new outcomes are discussed in the revised manuscript.

*# p1, line 15, should be "assess"*

This has been changed accordingly in the revised manuscript.

*# p1, line 16, "As with" might be better than "In similarity to"*

This has been changed accordingly in the revised manuscript.

*# p1, line 25, omit "," after storage*

This has been changed accordingly in the revised manuscript.

*# p2, line 23, replace ";" by ":"*

This has been changed accordingly in the revised manuscript.

*# p2, line 24, model not models*

This has been changed accordingly in the revised manuscript.

*# p2, line 27, omit "potentially"*

This has been changed accordingly in the revised manuscript.

*# p3, line 17, replace "lighter" by "less dense"*

This has been changed accordingly in the revised manuscript.

*# p3, line 23, replace "transport" by "trapping" (transport is controlled by permeability)*

This has been changed accordingly in the revised manuscript.

*# p4, line 9, replace "higher" by "greater"*

This has been changed accordingly in the revised manuscript.

*# p4, line 22, replace "catalases" by "cataclasites"*

This has been changed accordingly in the revised manuscript.

*# p4, line 24, insert "typically" before "> 1km"*

This has been changed accordingly in the revised manuscript.

*# p4, line 26, replace "with" by "in"*

This has been changed accordingly in the revised manuscript.

*# p5, line 24, insert "and" before "clay bed"*

This has been changed accordingly in the revised manuscript.

*# p5, line 24, replace "do" by "does"*

This has been changed accordingly in the revised manuscript.

*# p6, line 15, replace "sometimes" by "typically"*

This has been changed accordingly in the revised manuscript.

*# p7, line 30 onwards... I was confused by FRC (fault rock composition) and FRCC (fault rock clay content). I think you only need one of these, make it consistent in text and tables.*

This has been changed accordingly in the revised manuscript.

*# p9, line 3, insert "(see Figure 7)" after "57"*

This has been changed accordingly in the revised manuscript.

*# p9, line 26, insert "(and hence smaller pore-throat radius)" after "content" (see earlier comment)*

This has been changed accordingly in the revised manuscript.

*# Fig.4 - label the contours explicitly, to save the reader having to work out their values.*

This has been changed accordingly in the revised manuscript.

*# Table 1, last column. I assume the heading should be a lower case sigma, for standard deviation? The upper case symbol confused me for a while.*

This has been changed accordingly in the revised manuscript.

**R2**

*In your abstract your statement of your findings around line 15 is not as clearly stated as it could be – especially with regards to your finding that fault composition is important and it may be worth moving your sentence beginning "In contrast to hydrocarbon systems higher phyllosilicate. . .". To before your sentence beginning "However, the wettability of the carbon dioxide system is highly sensitive. .." only to*

*make your point that increasing phyllosilicate may not result in the expected increase in fault sealing for a co2 system.*

We have changed the abstract in the revised manuscript accordingly to the reviewer's suggestions.

*Page2 line 2 – while I agree that faults are ubiquitous in sedimentary basins, I would like to see a little more information provided on their extent, distribution, and scale, especially the impact of faults that are below seismic resolution.*

We thank the reviewer for their comment and have added additional information on the impact of subseismic-resolution faults to the revised manuscript as follows:

*"The scale and distribution of faults depends on the type of sedimentary basin and its geological history. In particularly faults that are below the resolution of seismic surveys cannot be avoided (Maerten et al., 2006; Le Gallo, 2016)."*

*Page 2 line 10 – faults can provide fault parallel flow for impermeable units as well? fig 1c if you are suggesting flow through the fracture – if you are not suggesting fault flow then you may nbeed to reword this section and figure.*

The reviewer points out an important point. Indeed faults can provide fault parallel flow through fracture networks within otherwise impermeable units. We have updated the text describing this in the revised manuscript as follows: *"…(iii) faults can provide fault parallel flow through fracture networks in otherwise impermeable rocks linking separate permeable units…"*.

Additionally, we have adjusted the caption of figure 1.

*Page 2 line 28 – "will accumulate underneath the flow barrier until breakthrough occurs due to the*
*increase in pressure within the reservoir" what do you mean by breakthrough? Capillary, spill point, induced fracturing? Can you be clearer with this statement?*

We have changed the statement in the revised manuscript accordingly as follows:

*"The fluid will accumulate underneath the flow barrier until capillary breakthrough or, less frequently, induced fracturing occurs due to the increase in pressure within the reservoir."*

*Page 3 line 30 – perhaps add the word seal before "rocks."*

This has been changed accordingly in the revised manuscript.

*Page 4 line 6 – you slightly contradict your statements at the end of page 3 where you infer fine grained rocks have small pore as page 4 states "Due to the heterogeneous nature of rocks the size of pores within*
*the sealing rock (fault rock or cap rock) varies"*

We have changed this sentence in the revised manuscript to highlight that the pore sizes within caprocks generally only varies slightly. In the revised manuscript it reads as follows:

*"Due to the heterogeneous nature of rocks the size of pores within the sealing rock (fault rock or cap rock) varies to a certain degree and thus two capillary pressures can be defined."*

*Page 8 line 5 – I would remind the reader that equations 5 to 9 are the three different author approaches (Bretan etc) as that is how you refer to them in the subsequent modelling discussions.*

This has been changed accordingly in the revised manuscript as follows:

*"Capillary threshold pressures for fault seals are calculated by using equations 5 to 9 (the algorithms by*
*Bretan et al. (2003), Yielding (2010) and Sperrevik et al. (2002)), these are then converted to the $CO_2$-brine system using equation 10, and subsequently column heights are calculated assuming a pore-throat size of 100 nm (eq. 3)."*

*Discussion section – I would like to see a clearer statement of your findings and a more logical layout to*
*your discussion as it is quite challenging to pick out the pertinent information rather than saying it "strongly influences" – I would like to see clearer statement of what he influence such as increasing phyllosilicate fault rocks results in one third less co2 stored - to make your findings clear from the start of this section which can then lead onto the detailed discussion of the reasons why.*

We thank the reviewer for their suggestion and have reworked the discussion section in the revised manuscript.